# PIX2STRUCT: SCREENSHOT PARSING AS PRETRAINING FOR VISUAL LANGUAGE UNDERSTANDING

## ABSTRACT

Visually-situated language is ubiquitous—sources range from textbooks with diagrams to web pages with images and tables, to mobile apps with buttons and forms. Perhaps due to this diversity, previous work has typically relied on domain-specific recipes with limited sharing of the underlying data, model architectures, and objectives. We present `Pix2Struct`, a pretrained image-to-text model for purely visual language understanding, which can be finetuned on tasks containing visually-situated language. `Pix2Struct` is pretrained by learning to parse masked screenshots of web pages into simplified HTML. The web, with its richness of visual elements cleanly reflected in the HTML structure, provides a large source of pretraining data well suited to the diversity of downstream tasks. Intuitively, this objective subsumes common pretraining signals such as OCR, language modeling, image captioning. In addition to the novel pretraining strategy, we introduce a variable-resolution input representation and a more flexible integration of language and vision inputs, where language prompts such as questions are rendered directly on top of the input image. For the first time, we show that a single pretrained model can achieve state-of-the-art results in six out of nine tasks across four domains: documents, illustrations, user interfaces, and natural images.

## 1 INTRODUCTION

Research on the interaction between language and vision has traditionally focused on tasks where images and text can be separated into distinct channels, e.g. visual question answering or image captioning. However, *visually-situated language* is a far more pervasive way in which these modalities interact and blend together. For example, documents, tables, infographics, and user interfaces (UIs) are intended to be consumed holistically, without clear boundaries between textual and visual elements (Figure 1). Comprehensive understanding of this information requires a deep set of skills, including the ability to recognize text, understand language, and incorporate diverse visual context.

Previous work on understanding visually-situated language is scattered. The focus is typically on complex task-specific combinations of available inputs and tools. For example, document-understanding models (Huang et al., 2022) rely on external OCR systems, UI-understanding models rely on platform-specific structural metadata (e.g. Android view hierarchy) (Bai et al., 2021), and diagram-understanding models rely on diagram parses (Kembhavi et al., 2016). Domain-specific engineering can be effective for high-resource settings such as documents, where there is an abundance of tools and data available. However, these pipelined models lack sharing of the underlying data, model architectures, and objectives across domains, limiting their general applicability. Moreover, relying on external systems like OCR increases engineering complexity, limits adaptability, and can increase overall computational cost. Recent work on OCR-free, end-to-end document understanding from images (Kim et al., 2022; Davis et al., 2022) has attempted to remove such task-specific engineering and reliance on external components during inference by learning to decode OCR outputs during pretraining—a significant step towards more general-purpose models. However, the focus on just text at the surface level limits the depth of knowledge transferred from unsupervised data. Effective use of pixel-only models remains an open challenge.

We present `Pix2Struct`, a pretrained model that combines the simplicity of purely pixel-level inputs with the generality and scalability provided by self-supervised pretraining from diverse and abundant web data. Specifically, we propose a *screenshot parsing* objective that requires predicting

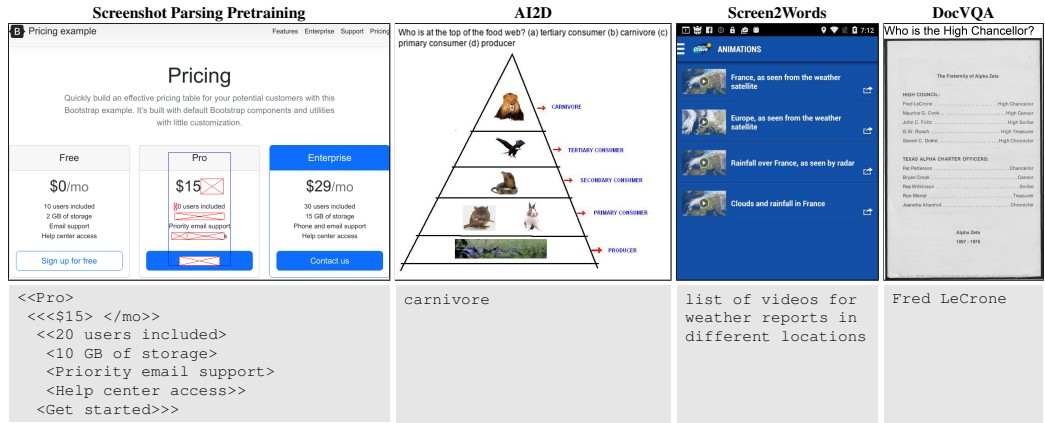

Figure 1: Examples of visually-situated language understanding tasks, including diagram QA (AI2D), app captioning (Screen2Words), and document QA (DocVQA). We also include an example of our proposed pretraining task (screenshot parsing) on the left. `Pix2Struct` directly encodes the pixels from the input image (above) and decodes the output text (below).

an HTML-based parse from a masked screenshot of a web page. HTML provides clean, vital signals about text, images, and layouts, while the masked inputs encourage joint reasoning about their co-occurrence. With the diversity and complexity of textual and visual elements found on the web, `Pix2Struct` learns rich representations of the underlying structure of web pages, which we show can effectively transfer to a variety of downstream visual language understanding tasks.

A key ingredient which enables this transfer is processing inputs visually and holistically as they are intended for human readers. We introduce variable-resolution inputs for vision transformers that prevent distortion of the original aspect ratio, which can vary greatly across documents, figures, and UIs. During finetuning, we render other inputs (e.g., questions in VQA and bounding boxes in UI tasks) onto the image input for the task. In effect, we consume all our inputs though a single modality, simplifying the modality combination problem in previous work.

We train two variants with 282M and 1.3B parameters, which we refer to as `Pix2Struct`-Base and `Pix2Struct`-Large respectively, on 80M screenshots of web pages from the C4 corpus (Raffel et al., 2020). Experiments on four domains and nine tasks show that our finetuned models strongly outperform Donut (ranging from 9 to 53 points), the strongest existing baseline without pipelines. Compared with baselines with domain-specific pipelines, we lag behind the state of the art in high-resource domains such as documents and natural images, but we observe significant improvements (ranging from 1 to 44 points) in low-resource domains such as illustrations and UIs. We hope that these results encourage the community to continue developing such general-purpose methods and further enable new applications in this currently fragmented intersection of language and vision.

To summarize, our major contributions are as follows:

- We introduce the area of general-purpose visually-situated language understanding, which consists of diverse tasks but common challenges.

- We propose a *screenshot parsing* pretraining objective based on the HTML source of web pages. We show that our objective is more effective than previous attempts at enabling the elegant pixel-to-text design for general-purpose visually-situated language understanding.

- We introduce variable-resolution input representations to the Vision Transformer and new fine-tuning strategies that seamlessly integrate language and vision inputs by directly rendering any language prompts on top of the input image.

- The pretrained checkpoints and code for reproducing results for all nine tasks are available at `https://github.com/anonymized/pix2struct`.

## 2 METHOD

### 2.1 BACKGROUND

Prior attempts at pixel-only modeling of visually situated language have largely focused on documents and natural images. For documents, Donut (Kim et al., 2022) and Dessurt (Davis et al., 2022) combine pretrained objectives based on surface-level features from synthetic images or predicted OCR outputs. For natural images, concurrent work—GIT2 (Wang et al., 2022a) and PaLI (Chen et al., 2022b)—focus on collecting and training on large scale image captioning data that transfers well to datasets with natural images (e.g. TextCaps).

We aim to provide a single pretrained model that can be finetuned on a wider variety of tasks and domains. The input to our model is an image in the form of raw pixels only, and the output of our model is text in the form of token sequences, similar to Donut. The goal is a visual analog of models like T5 (Raffel et al., 2020), where the generality of simple inputs and outputs is combined with the power of pretraining on large unsupervised sources of data. During finetuning, the complexity of adapting to diverse downstream tasks resides only in data preprocessing.

Even without visual context, pixel-only language modeling for text has only recently been attempted (Rust et al., 2022)—perhaps because it requires solving multiple hard sub-problems. First, the ability to read with high fidelity while at the same time building rich high-level representations poses a difficult optimization problem. Second, encoding text-heavy inputs (e.g. long documents) involves processing high-resolution images with variable aspect ratios. State-of-the-art document understanding models (Huang et al., 2022) therefore rely on the combination of (possibly noisy) OCR outputs with low resolution images. We argue that the reliance on OCR has prevented exploration of learning more general-purpose representations that meaningfully extend beyond the text.

We show the various components of `Pix2Struct` that address these challenges. Section 2.2 discusses modifications to the transformer inputs to handle variable aspect ratios and resolutions. We then discuss our proposed screenshot parsing objective (Section 2.3) and how curriculum learning leads to more robust transfer learning (Section 2.4). Finally, Section 2.5 shows how `Pix2Struct` consumes textual and visual inputs (e.g. questions and images) in the same space by rendering text inputs onto images during finetuning.

### 2.2 ARCHITECTURE

`Pix2Struct` is an image-encoder-text-decoder based on the Vision Transformer (ViT) (Dosovitskiy et al., 2021). While the bulk of the model is fairly standard, we propose one small but impactful change to the input representation to make `Pix2Struct` more robust to various forms of visually-situated language. Before extracting fixed-size patches, the standard ViT scales the input images to a predefined resolution, which creates two undesirable effects: (1) rescaling the image distorts the true aspect ratio, which can be highly variable for documents, mobile UIs, and figures. (2) transferring these models to downstream tasks with higher resolution is non-trivial (Touvron et al., 2019; Wang et al., 2021b), since the model only observes one specific resolution during pretraining.

We instead propose to always scale our input image up or down such that we extract the maximal number of patches that fit within the given sequence length (see Figure 5 in Appendix B). In order for the model to handle variable resolutions unambiguously, we use 2-dimensional absolute positional embeddings for the input patches. Together these changes to the standard ViT inputs provide two major advantages in terms of robustness to: (1) extreme aspect ratios, which is common in the domains that we experiment with, and (2) on-the-fly changes to the sequence length and resolution.

### 2.3 PRETRAINING

The goal of pretraining is for `Pix2Struct` to represent the underlying structure of the input image. To that end, we create self-supervised pairs of input images and target text from a web corpus. For each web page in the pretraining corpus, we start by collecting its screenshot and HTML source.

**Screenshot parsing inputs & outputs**   The screenshot and HTML are modified to ensure rich and dense learning signal during pretraining. These modifications provide a reasonable trade-off between preserving the semantics of the page and requiring a practical decoder sequence length.

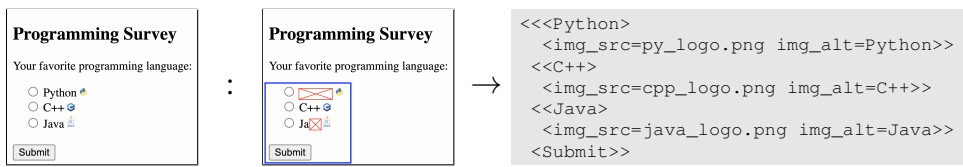

Figure 2: Toy illustration of input-output pairs (right) sampled from the original web page (left).

We condense the HTML DOM tree by (1) only keeping nodes with visible elements or descendants with visible elements and (2) if a node does not contain visible elements and it only has a single child, replacing the singleton child with any grandchildren to remove chained nesting. In each node, we only keep information about text and images, as represented by their filename and any alternative text. Much more information could be retained, such as element tags, style, bounding boxes, titles and URLs etc in future work. The decoder sequence length is further reduced by finding the largest subtree that fits within a predefined sequence length when linearized. A bounding box indicating the region covered by the chosen subtree is also drawn on the screenshot.

For better context modeling, we introduce a BART-like (Lewis et al., 2020) learning signal by masking 50% of the text while decoding the entire subtree. The masked regions are randomly sampled spans of text from the chosen subtree where we draw crossed-out opaque bounding boxes (Figure 2).

**Comparison to existing pretraining strategies** Our proposed screenshot parsing seamlessly integrates signals reminiscent of several well-known pretraining strategies:

- Recovering the unmasked parts of the parse is similar to OCR, a prerequisite skill for understanding language. OCR pretraining was proposed in Donut which uses synthetic renderings or predicted OCR outputs. In Figure 2, predicting `<C++>` exemplifies this learning signal.
- Recovering the masked parts of the parse is much like masked language modeling (Devlin et al., 2019). A major difference is that the visual context often provides additional cues useful for sharpening the predictions. In Figure 2, predicting `<Python>` exemplifies this learning signal.
- Recovering the alt-text from images is a common pretraining strategy for image captioning (Sharma et al., 2018; Wang et al., 2022a; Chen et al., 2022b). A major difference is that the model is permitted to use the web page as additional context. In Figure 2, predicting `img_alt=C++` exemplifies this learning signal.

Appendix F contains more examples of screenshots paired with their gold and predicted parses.

## 2.4 WARMING UP WITH A READING CURRICULUM

While we can directly pretrain `Pix2Struct` on the screenshot parsing task, we find that doing this naively can result in instability and slow learning. However, if we first expose the model to a short, intense "warmup" stage of simply learning to read, we find a strong curriculum learning effect where (1) pretraining is more stable and converges faster, and (2) we observe better finetuning performance, as discussed in Section 5. Specifically, we create images of text snippets with random colors and fonts on a white background. The model simply needs to decode the original text (see Appendix E for an example). This type of curriculum learning was also used in Dessurt (Davis et al., 2022) and can also be viewed as a simplified version of Donut's pretraining.

## 2.5 FINETUNING

Finetuning `Pix2Struct` is straightforward and largely a matter of preprocessing the downstream data to unambiguously reflect the task in the image inputs and text outputs, analogous to the way T5 (Raffel et al., 2020) is used for text-based tasks. In this section, we cover the preprocessing strategies for the tasks described in Table 2. Examples of this preprocessing are shown in Figure 1.

Captioning is the most straightforward, since the input image and the output text can be directly used (as in TextCaps, Screen2Words). In the case where the focus of the caption is a specific bounding box (as in Widget Captioning), we draw the target bounding box on the image itself.

For visual question answering (as in OCR-VQA, ChartQA, DocVQA, InfographicsVQA), while multimodal models typically reserve a specialized text channel for the question, we opt to instead directly render the question as a header at the top of the original image. `Pix2Struct` reads both the question and the image jointly via the visual modality. This strategy is analogous to the common practice of simply concatenating all inputs during finetuning of pretrained text models, first proposed in GPT (Radford et al., 2018) and has been the default method in NLP since then. Intuitively, this strategy is effective because `Pix2Struct` has been pretrained to be sensitive to long-range interactions between various parts of the input image. In the case of multiple choice answers (as in AI2D), we also render the choices in the header as part of the question.

The most complex scenario is RefExp, where the task is choosing between UI components that a natural language expression could be referring to. For each candidate, we create a training instance where the input image contains the bounding box and referring expression, and the decoding target is "true" or "false". We sample five negative candidates per positive candidate during training. During inference, we pick the candidate for which the model generates "true" with the highest score.[1]

## 3 Experimental Setup

### 3.1 Benchmarks

We evaluate `Pix2Struct` on multiple benchmarks centered around visually-situated language understanding across four domains: illustrations, user interfaces, natural images, and documents. Since we are the first to aggregate datasets with this scope, we optimized for diversity in domains as well as in task-format. Evaluation is restricted to standard splits without additional labeled data. Table 2 provides a summary of the datasets with details discussed in Section 4.

We use evaluation metrics as defined in the original papers: (a) average normalized Levenshtein similarity (ANLS) for DocVQA and InfographicVQA, (b) exact match (EM) for AI2D, RefExp, and OCR-VQA, (c) relaxed accuracy (RA) for ChartQA, and (d) CIDEr for the generation tasks.

### 3.2 Implementation and Baselines

**Pretraining** We pretrain two model variants: (a) a *base* model with 282M parameters including 12 transformer layers with a hidden size of 768, and (b) a *large* model with 1.3B parameters including 18 layers with a hidden size of 1536. Both models have the same warmup stage using text rendered from BooksCorpus (Zhu et al., 2015) lasting 30K steps with a maximum input sequence length of 128 patches. The base model is then pretrained further for 270K steps with the screenshot parsing objective using a batch size of 3072 on 64 Google Cloud TPUs. The large model is pretrained for 170K steps with a batch size of 1024 on 128 Google Cloud TPUs. Both models use an input sequence length of 2048 patches and are optimized using Adafactor (Shazeer & Stern, 2018). The learning rate schedule uses a linear warmup of 1000 steps to 0.01, followed by cosine decay to 0. The decoder sequence length is 128 tokens, and we choose pretraining targets to have at most 1024 characters. As a reference point, the base model reaches  30 BLEU and the large model reaches  32 BLEU. Details about finetuning can be found in Appendix D.

**Baselines** Across all tasks, we found an exceedingly large number of methods which could serve as baselines. We compare our results against state of the art (SotA) methods in each domain (see Section 4 for method descriptions). Several methods use model ensembles, multitask with labeled training data from other datasets (Powalski et al., 2021; Wang et al., 2022a), or use validation data for training (Li et al., 2021a). For fair comparison and ease of experimentation, we focus on single-model and single-task baselines trained on standard splits. Several (per-task) SotA (Li et al., 2021c; Masry et al., 2022) use domain-specific inputs (e.g. view hierarchies for UIs or gold data tables for charts) making it difficult to apply them to other domains. For a strong, consistent visual baseline across domains, we finetuned Donut on all tasks where a purely visual baseline was unavailable.[2]

---

[1]or the lowest score if something other than "true" was generated (almost always "false".)

[2]Except RefExp due to the complexity of required inference modifications.

| Method | Pretraining | Chart QA | AI2D | OCR VQA | Ref Exp | Widget Cap | Screen2 Words | Text Caps | Doc VQA | Info VQA |
|---|---|---|---|---|---|---|---|---|---|---|
| State of the art w/ pipelines | - | (VTP) 45.5 | (DQAN) 38.5 | (LATr) 67.5 | (UIB) 90.8 | (VUT) 97.0 | (VUT) 64.3 | (PaLI) **160.4** | (LLMv3) **83.4** | (T52DU) **46.1** |
| GIT2 | Image captioning | - | - | 70.3 | - | - | - | 145.0 | - | - |
| Donut | OCR | 41.8 | 30.8 | 66.0 | - | 127.4 | 56.4 | 74.4 | 67.5 | 11.6 |
| Pix2Struct | | | | | | | | | | |
|   Base | Screenshot parsing | 56.0 | 40.9 | 69.4 | 92.2 | 133.1 | 107.0 | 88.0 | 72.1 | 38.2 |
|   Large | Screenshot parsing | **58.6** | **42.1** | **71.3** | **94.2** | **136.7** | **109.4** | 95.5 | 76.6 | 40.0 |

(Left margin vertical label: Pixel only)

Table 1: `Pix2Struct` outperforms prior visual methods on 8 out of 9 benchmarks with SotA results on 6. While GIT2's image captioning-based pretraining understandably helps on TextCaps, our screenshot parsing objective transfers to a wider variety of downstream tasks. The individual pipeline SotA methods are described in Section 4 with full results in Appendix C.

## 4 RESULTS

We discuss here the main results comparing `Pix2Struct` with prior work, as shown in Table 1.

### 4.1 ILLUSTRATIONS

**ChartQA** (Masry et al., 2022) is a VQA dataset with questions based on charts, i.e. visual representations of tabular data.[3]. VisionTaPas (Masry et al., 2022), the current SotA, is a pipeline which operates on data tables predicted from the given charts. It consists of (1) a vision transformer encoder for encoding the chart image, (2) a TaPas encoder for encoding the question and the data table, and (3) a cross-modal encoder. In contrast, `Pix2Struct` does not rely on noisy table extractors and uses the given chart directly—improving the SotA from 45.5 to 58.6 with the large variant.

**AI2D** (Kembhavi et al., 2016) contains multiple choice questions based on illustrative science diagrams (about geological processes, biological structures etc.). The dataset comes with only train and test splits. We set aside 1% of the train split for validation. The current SotA DQA-NET (Kembhavi et al., 2016) focuses on modeling entity relationships via a pipeline of tools for extracting arrows, blobs, and other visual elements. `Pix2Struct`-Large outperforms DQA-NET and Donut by 3.6 and 11.27 points respectively without any domain-specific modifications.

**OCR-VQA** (Mishra et al., 2019) is a VQA dataset on images of book covers. The questions are based on book metadata such as title, author, genre etc. Much of work on OCR-VQA, including the pipeline SotA LATr (Biten et al., 2022), uses off-the-shelf OCR. Concurrent work, GIT2 (Wang et al., 2022a), the current SotA, is pretrained on 12.9B image caption pairs. Their final finetuning stage is preceded by intermediate finetuning on eight VQA datasets including VQAv2 (Goyal et al., 2017), VizWiz-VQA (Chen et al., 2022a), and OCR-VQA (Mishra et al., 2019) amongst others. Despite not using more labeled training data, we outperform GIT2 by almost 1 point.

### 4.2 UIS

**RefExp** (Bai et al., 2021) Given a natural language referring expression, an app screenshot, and a set of components (via bounding boxes on the screenshot), the goal here is to retrieve the component that the expression refers to. UIBert (Bai et al., 2021), the current SotA, is pretrained on a combination of inputs from mobile apps including screenshots, OCR text, and Android view hierarchies. Our models substantially ourperform UI Bert by 1.4 and 3.4% absolute, with `Pix2Struct`-Large setting the new state of the art with 94% accuracy.

**Widget Captioning** (Li et al., 2020b) is an image captioning task where the input is an app screenshot annotated with a single bounding box denoting a widget (e.g. a button or a scroll bar). The caption describes the functionality of the widget (e.g. *find location*). VUT (Li et al., 2021c), the current SotA uses a specialized UI encoder combining images, bounding boxes, and view hierarchies. `Pix2Struct`-Large improves the state of the art CIDEr from 97.0 to 136.7.

---

[3]We evaluate on the task without the gold data table

**Screen2Words** (Wang et al., 2021a) is an image captioning task where the input is an app screenshot and the caption describes the functionality of the page (see Figure 1 for an example). `Pix2Struct`-Large improves the state of the art CIDEr from 64.3 to 109.4.

### 4.3 NATURAL IMAGES

**TextCaps** Concurrent with our work, GIT2 (5.1B parameters) and PaLI (17B parameters) have advanced the state of the art on TextCaps by pretraining on 10B+ image-caption pairs extracted from the web. PaLI (CIDEr 135.4 without OCR) and GIT2 (CIDEr 145) show comparable performance when finetuned without OCR based inputs. PaLI achieves SotA (CIDEr 160.4) performance when finetuned with OCR, indicating that even large-scale methods, end-to-end pixel-only performance lags behind pipeline SotA. While their image captioning-based pretraining understandably helps on TextCaps, previous work (Kim et al., 2022) shows that captioning does not necessarily transfer to other domains like documents. Moreover, screenshot parsing subsumes signals from image captioning (Section 2.3) while using a fraction of the the data used for pretraining GIT2 and PaLI. Overall, these results indicate that `Pix2Struct` could benefit from scaling even further in future work.

### 4.4 DOCUMENTS

**DocVQA** (Mathew et al., 2021) is a dataset of questions about scanned documents,[4] include type-written, printed, handwritten and born-digital text. `Pix2Struct`-Large outperforms Donut, the previous visual SotA on DocVQA by 9 points. Top-performing single-task methods like LayoutLMv3 (Huang et al., 2022) (ANLS 83.4) typically use three components: (a) an off-the-shelf OCR system, (b) pretrained text and image encoders, and (c) additional pretraining on the IIT-CDIP scanned documents corpus. Despite using purely visual representations and no in-domain pretraining data, `Pix2Struct` achieves competitive performance (ANLS 76.6).

**InfographicVQA** (Mathew et al., 2022) is a dataset of questions about infographics from the web. A unique challenge of this dataset is its large images with extreme aspect ratios. Donut scales images to a fixed aspect ratio, which we speculate is the cause of its poor performance with an ANLS of 11.6. `Pix2Struct`-Large sets the state of the art amongst visual models with an ANLS of 40.

For both DocVQA and InfographicVQA, text-only baselines are at or near the state of the art. A T5-based model (T5 + 2D + U) with 2D positional biases (Borchmann et al., 2021) achieves ANLS of 81 on DocVQA and a SotA ANLS of 46.1 on InfographicVQA. This is in part due to the text-heavy nature of the data (especially DocVQA) where visual context plays a lesser role, and the more mature pretrained text-based encoders can do the heavy lifting.

**Common trends** Overall, `Pix2Struct` outperforms Donut in all tasks underscoring the effectiveness of our pretraining. We also advance the single-task state of the art on six out of nine benchmarks across four domains. Scaling up from base to large results in considerable improvements on all tasks despite the base model making 4.5 times as many iterations over the data compared to the large version. Results indicate that further scaling up of `Pix2Struct` is a promising direction.

## 5 ANALYSIS

**Ablating pretraining objectives** Table 3 analyzes the importance of each component of our pretraining recipe on DocVQA, Widget Captioning, and TextCaps validation sets. The full pretraining recipe consists of a warmup reading stage on the books corpus followed by pretraining using the screenshot parsing objective. For these experiments, we use the base variant with a total of 100K steps of pretraining including 30K warmup steps followed by 70K steps of screenshot parsing. The screenshot parsing ablation removes the screenshot parsing stage altogether and uses an extended warmup stage of 100K steps. The warmup ablation removes the warmup stage and pretrains the next stage (from random initialization) for 100K steps. The masking ablation uses 30K steps warmup (like the full model) followed by 70K steps of screenshot parsing without masking.[5]

---

[4]from the UCSF Industry Documents Library `https://www.industrydocuments.ucsf.edu`
[5]All models use the same hyperparameters, including a batch size of 3072.

| Pretraining | Doc VQA | Widget Captioning | Text Caps |
|---|---|---|---|
| Full | 67.8 | 137.5 | 84.2 |
| – Warmup | 56.2 | 128.0 | 71.7 |
| – Masking | 55.7 | 129.4 | 77.4 |
| – Screenshot Parsing | 12.2 | 35.1 | 24.2 |

Figure 3: Ablations of pretraining components. Each ablation is a modification with respect to the full model, while keeping the total number of pretraining steps constant.

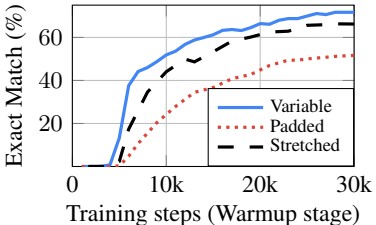

Figure 4: Our variable-resolution inputs prevent aspect-ratio distortion while minimizing padding.

Table 3 shows that all components of the pretraining scheme are crucial for good downstream task performance across all benchmarks. The biggest drop in performance comes from ablating the screenshot parsing stage, effectively reducing the pretraining to reading linear text. Ablating the warmup and masking is nearly equivalent on DocVQA and Widget Captioning while the warmup is slightly more important in TextCaps. Overall, our results seem to indicate that reading and understanding visually-situated language is a complex problem which needs a deep set of skills including recognizing text, understanding language, and incorporating visual context.

**Ablating variable-resolution inputs** Figure 4 compares various ways to convert input images into a constant number of patches. This ablation is performed on the warmup stage (Section 2.4), where we measure full sequence accuracy. The 'padded' variant maintains the original aspect ratio, but introduces significant padding, which sacrifices the effective resolution. The 'stretched' variant, typically used in ViT, introduces no padding but distorts the original image. Our variable-resolution inputs get the best of both worlds by maintaining the original aspect ratio while maximizing the budget specified by the sequence length.[6] Experiments show that this benefit leads to more effective learning, even for a task as simple as transcribing text in the input image.

## 6 DISCUSSION

In this section, we lay out some of the challenges in training general-purpose visual language understanding models, and discuss a road map for future work.

**Resolution** Like Donut, we found that pretraining and finetuning performance are extremely sensitive to the input resolutions.[7] The difficulty in using high-resolution images has been a bottleneck for pixel-only models since higher resolutions often lead to longer sequence lengths. This bottleneck has in part been responsible for the dominance of OCR-based pipelines which are able to use lower image resolutions due to a dedicated text encoder.[8] However, steady progress with Donut and `Pix2Struct` combined with recent progress in long range transformers (Press et al., 2021) provides hope that pixel-only models will bridge the gap with OCR-based pipelines.

**The visual web** As a first attempt towards a general-purpose visual language understanding model, we focused on simplicity both in terms of how we use the HTML source and our choice for the pretraining corpus, C4—a known public corpus used in previous work (Raffel et al., 2020) that is significantly smaller and narrower than corpora used to train the largest language models today. However, web data includes even richer multimodal signals such as videos and interactions. We posit that future versions of general-purpose visual language understanding models will benefit from better data curation. This opportunity also comes with a caveat: just like text-based models, we must be careful of harmful content on the web, which multimodal models would also be sensitive to.

**Generality** While we have focused on general pixel-only models, we do acknowledge that using OCR-pipelines or metadata can be appropriate or even necessary in certain domains. For NLP, the scaling of pretrained text based models has led to not only simpler model architectures and preprocessing, but also emergent abilities on newer tasks which were hitherto considered far too

---

[6]Illustrations of our variable-resolution method and the standard stretching method is in Appendix B.

[7]See Appendix B for a concrete comparison.

[8]OCR pipelines, while noisy, often result in manageable sequence lengths for large-scale text encoders.

difficult (Wei et al., 2022). A general-purpose model may also enable broader applications for visual language, e.g. filling in missing accessibility annotations (Zhang et al., 2021). The broader objective of this work is to bring pretraining for visually-situated language understanding a step closer to text-base counterparts and pave the way for similar benefits from data and model scaling.

# 7 RELATED WORK

To the best of our knowledge, no prior work has pretrained and evaluated a visually-situated language understanding model on tasks spanning all four domains of documents, illustrations, user interfaces, and natural images. [9] We build on prior work primarily focused on a single domain and briefly highlight the similarities as well as the points of departure with respect to such work here.

**Document understanding** State-of-the-art models in this domain are based on a pipeline of an external OCR system and a model that combines images and OCR annotations (Appalaraju et al., 2021; Powalski et al., 2021; Xu et al., 2021), *inter alia*. Prominent representatives are LayoutLMv3 (Huang et al., 2022), which uses a simplified Transformer-based architecture and losses that encourage patch–OCR alignment. TILT (Powalski et al., 2021) pretrains a text decoder and an image + OCR-output encoder followed by intermediate finetuning on multiple QA tasks. `Pix2Struct` is more closely related to Donut and Dessurt (Davis et al., 2022), also image-to-text models without OCR at inference time; the main difference stems from our more powerful pretraining task from ground truth structures and resolution flexibility enabling transfer to a variety of visual language domains.

**UI understanding** Models in this group have focused solely on the UI domain and have been pretrained on data from mobile and web apps. While some models use image-only inputs (Liu et al., 2018; Chen et al., 2020), higher accuracy approaches also make use of the structures of view hierarchies and element annotations, e.g. UIBert (Bai et al., 2021), ActionBert (He et al., 2021), VUT (Li et al., 2021c) although the structured metadata is known to be noisy (Li et al., 2020a). The screen parsing task (Wu et al., 2021), while similar in name, is an amalgamation of pipelines over domain-specific structures that are not intended to produce transferable representations.

**Natural image understanding** A variety of model architectures (Singh et al., 2019; Sidorov et al., 2020; Wang et al., 2020) and objectives (Yang et al., 2021) have been proposed for understanding natural images containing short segments of text (e.g. street signs). The predominant source of pretraining data has been image-caption pairs often in conjunction with the output of OCR (Chen et al., 2022b; Yang et al., 2021). GIT2 (Wang et al., 2022a), the pixel-only SoTA, learns from 12.9 billion image-caption pairs and is about 4 times larger than `Pix2Struct`— it outperforms our model significantly on natural images (TextCaps) but underperforms on illustrations (OCR-VQA). PaLI benefits from using a pipeline with OCR, obtaining higher performance on TextCaps. These methods have not been evaluated on more text-dense input domains.

**Illustrations** Models for illustrations have not been fully pretrained on large scale data, perhaps because such data is not readily available. Some components of such models, e.g. T5 and TaPas (Eisenschlos et al., 2020) used in the VL-T5 and VisionTaPas models of Masry et al. (2022) or LATr's OCR output encoder (Biten et al., 2022) have been pretrained on digital-born or OCR-ed documents. Our approach outperforms current SotA models, without relying on other intermediate structures.

**Related models learning from markup structure** MarkupLM (Li et al., 2021b) and Webformer (Wang et al., 2022b) learn encoders of HTML from web pages. HTLM (Aghajanyan et al., 2021) and CM3 (Aghajanyan et al., 2022) are generative models of simplified HTML to enable powerful zero-shot prompting with text and natural images. Im2Tex (Deng et al., 2017) is conceptually the most relevant in showing that a pixel-only parser can be learned from freely-available pairs of markup and renders, but they do not focus on transferring this signal to wider applications.

**Datasets** We have selected a set of datasets representing challenges in visually-situated language understanding in a variety of domains, but our selection is not aimed to be exhaustive. The DUE benchmark (Borchmann et al., 2021) focuses on a more limited domain of visual document understanding (e.g. excluding natural images and UIs), but integrates a more comprehensive set of tasks within that domain, including document layout analysis and academic papers.

---

[9]Some prior approaches have been evaluated on two of these domains.

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

## A    DATASET DETAILS

| Dataset | Domain | Description |
|---|---|---|
| OCR-VQA | Illustrations | VQA over book covers. |
| ChartQA | Illustrations | VQA over charts (visualization of tabular data) |
| AI2D | Illustrations | VQA over science diagrams |
| RefExp | UIs | Detect UI component matching a natural language query |
| Widget Captioning | UIs | Captioning a UI component on a screen |
| Screen2Words | UIs | Captioning a UI screen to describe functionality |
| TextCaps | Natural images | Captioning of natural images containing text |
| DocVQA | Documents | VQA over scanned documents. |
| InfographicsVQA | Documents | VQA over high-res infographics. |

Table 2: Summary our proposed diverse benchmark for visually-situated language understanding

## B    RESOLUTION IN VISUALLY-SITUATED LANGUAGE UNDERSTANDING TASKS

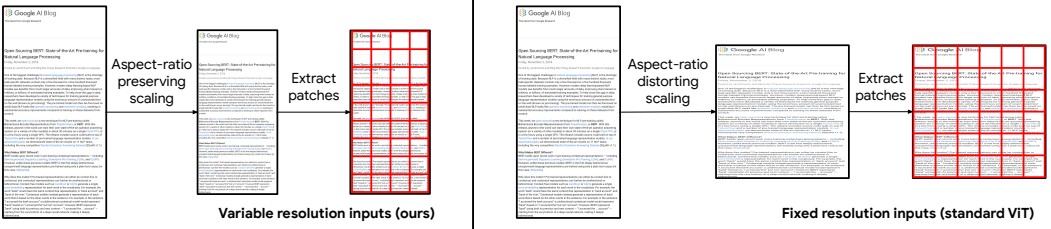

Figure 5: Comparison of our variable resolution inputs and the typical fixed resolution input. We illustrate the preprocessing for a target sequence length of 36 patches for both inputs.

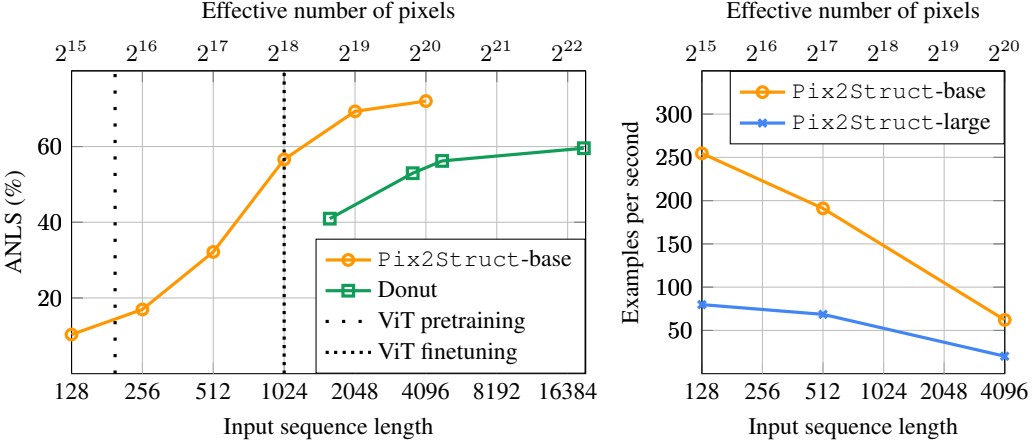

Figure 6: Overview of the impact of resolution on the DocVQA task. Note that the bottom axis only applies to Pix2Struct. Pix2Struct is also the only model that adapts to various resolutions seamlessly, without any retraining or post-hoc parameter creation. (Left) In both Donut and Pix2Struct, we show clear benefits from use larger resolutions. (Right) Inference speed measured by auto-regressive decoding (max decoding length of 32 tokens) on the validation set of DocVQA using a v3-8 Cloud TPU.

Previous methods rescale input images to fixed resolutions, which can introduce aspect ratio distortions that can be severe for inputs such as webpages and documents. In contrast, we prevent aspect ratio distortion by rescaling input images up or down such that we extract the maximal number of patches that fit within the given sequence length (Figure 5).

Figure 6 gives an overview of the importance of input resolutions in visually-situated language understanding tasks. Though `Pix2Struct` is more efficient at making use of the input resolution, both `Pix2Struct` and Donut require high resolutions to perform well on DocVQA (note the log scale). For example, we only see significantly diminishing returns after about 1M pixels (4096 patches of $16 \times 16$ pixels for `Pix2Struct` and $1024 \times 1024$ for fixed-resolution models). However, ViT models typically pretrain with resolutions of $224 \times 224$ and finetune with up to $512 \times 512$. This is a subtle but critical detail that makes using standard ViT out of the box suboptimal.

On the right of Figure 6, we also present example inference speeds on a v3-8 Cloud TPU when performing inference on DocVQA. At full resolution (4096 sequence length or 1M pixels), the base model processes 62 documents per second, and the large model processes 20 documents per second.

## C  FULL RESULTS

| | Method | Chart QA | AI2D | OCR VQA | Ref Exp | Widget Cap | Screen2 Words | Text Caps | Doc VQA | Info VQA |
|---|---|---|---|---|---|---|---|---|---|---|
| Pipelined | TILT | - | - | - | - | - | - | - | 87.1* | - |
| | VUT | - | - | - | - | 94.8 | 64.3 | - | - | - |
| | TAP | - | - | - | - | - | - | 99.5 | - | - |
| | LATr | - | - | 67.5 | - | - | - | - | - | - |
| | PLC | - | - | - | - | 97.0 | - | - | - | - |
| | T5 + 2D + U | - | - | - | - | - | - | - | 81.0 | **46.1** |
| | RoBERTa | - | - | - | - | - | - | - | 69.5 | - |
| | LayoutLMv3 | - | - | - | - | - | - | - | **83.4** | - |
| | DQA-NET | - | 38.5 | - | - | - | - | - | - | - |
| | UI Bert | - | - | - | 90.8 | - | - | - | - | - |
| | M4C | - | - | 63.9 | - | - | - | 81 | - | 14.7 |
| | VisionTaPas | 45.5 | - | - | - | - | - | - | - | - |
| | PaLI | - | - | - | - | - | - | **160.4** | - | - |
| Pixel only | GIT2 | - | - | 70.3* | - | - | - | 145.0 | - | - |
| | Donut | 41.8 | 30.8 | 66.0 | - | 127.4 | 56.4 | 74.4 | 67.5 | 11.6 |
| | `Pix2Struct`-Base | 56.0 | 40.9 | 69.4 | 92.2 | 133.1 | 107.0 | 88.0 | 72.1 | 38.2 |
| | `Pix2Struct`-Large | **58.6** | **42.1** | **71.3** | **94.2** | **136.7** | **109.4** | 95.5 | 76.6 | 40.0 |

Table 3: Amongst single-task single-model methods, `Pix2Struct` achieves state-of-the-art results on 6 out of 9 benchmarks spanning 4 domains. * indicates that the method used additional labeled data from other tasks and are not directly comparable to single task methods. VisionTaPas uses a table extraction tool. DQA-NET uses diagram processing tools for detecting arrows, blobs, etc in addition to standard OCR. UI Bert and VUT use Android view hierarchies. All other non-image methods use standard OCR.

Table 3 reports full results for pipeline and pixel-only methods across all datasets. For fair comparison and ease of experimentation, we focus on single-model and single-task baselines trained on standard splits. Several (per-task) SotA (Li et al., 2021c; Masry et al., 2022) use domain-specific inputs (e.g. view hierarchies for UIs or gold data tables for charts) making it difficult to apply them to other domains.

## D  HYPERPARAMETERS

**Finetuning**  The base and large models are finetuned with an input sequence length of 4096 and 3072 respectively, except the base model on InfographicVQA which benefits from a longer sequence length of 6144. We cannot use a longer sequence length for the large variant due to TPU/GPU memory constraints. We finetune for 5000 or 10000 steps with a batch size of 32, 128, or 256, with hyperparameter tuning and early stopping based on the validation set. Table 4 contains hyperparameter values for all tasks.

| Dataset | Base | | | Large | | |
|---|---|---|---|---|---|---|
| | Seq Len | Batch | Steps | Seq Len | Batch | Steps |
| DocVQA | 4096 | 256 | 10000 | 3072 | 128 | 10000 |
| InfographicVQA | 6144 | 64 | 10000 | 3072 | 128 | 10000 |
| AI2D | 4096 | 32 | 5000 | 3072 | 32 | 5000 |
| ChartQA | 4096 | 256 | 10000 | 3072 | 128 | 10000 |
| OCR-VQA | 4096 | 256 | 10000 | 3072 | 128 | 10000 |
| RefExp | 4096 | 256 | 10000 | 3072 | 128 | 10000 |
| Screen2Words | 4096 | 32 | 10000 | 3072 | 32 | 10000 |
| Widget Cap. | 4096 | 256 | 5000 | 3072 | 128 | 5000 |
| TextCaps | 4096 | 256 | 5000 | 3072 | 128 | 5000 |

Table 4: Model hyperparameters

# E    WARMUP STAGE EXAMPLE

Figure 7: Example of input-output pairs during the warmup stage.

Exposing the model to a short, intense "warmup" stage of simply learning to read, results in a strong curriculum learning effect where (1) pretraining is more stable and converges faster, and (2) we observe better finetuning performance. Figure 7 shows an example of rendered text from the Books corpus with its "parse".

# F    PRETRAINING EXAMPLES

The figures below show screenshots of our pretraining data along with ground-truth and predicted parses.

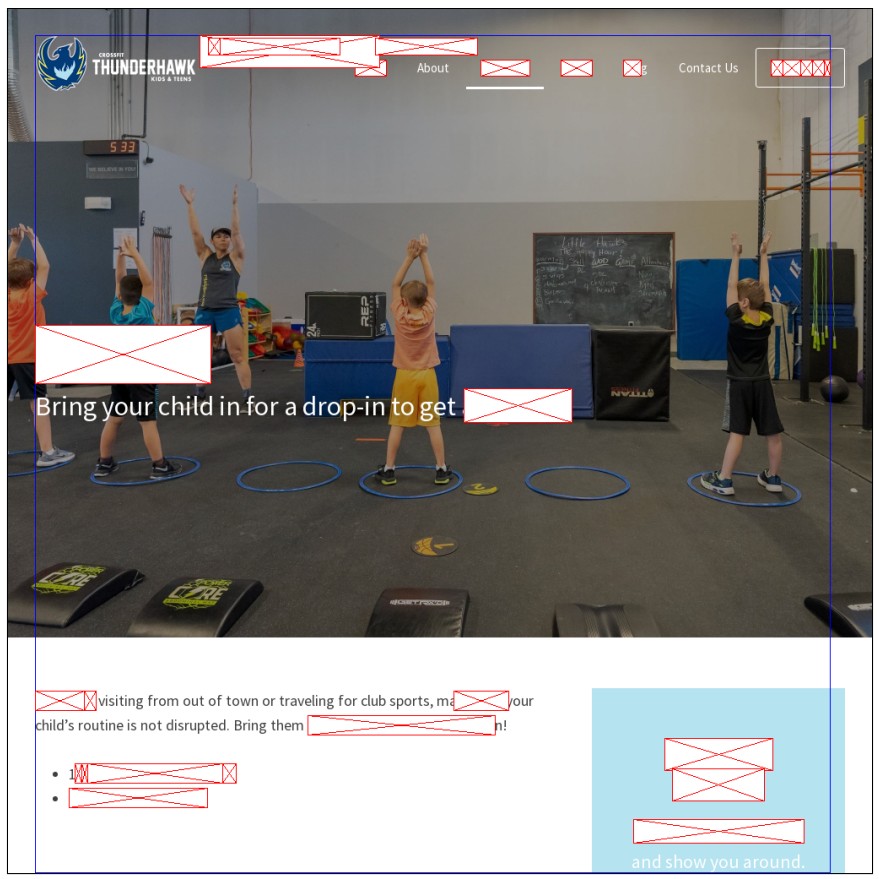

**Ground-truth Parse**

```
<<<<CrossFit Thunderhawk | Rio Rancho>
   <dedicated to promote healthy kids and teens in Rio Rancho, NM>>
  <<Home> <About> <Schedule> <Media> <Blog> <Contact Us> <Free Class>>>
 <<Drop-ins>
  <Bring your child in for a drop-in to get a WOD in!>>
 <<<If you are visiting from out of town or traveling for club sports,
   make sure your child's routine is not disrupted. Bring them in for
   a drop in to get a WOD in!>
  <<1-day CrossFit Athlete $15>
   <1-day Competitor $25>>>
  <<Become A Member>
   <We'd love to meet you and show you around.>>>>
```

**Predicted Parse**

```
<<<<img_src=thunderhawk-logo-white img_alt=Thunderhawk Sports & Fitness>
   <Thunderhawk Sports & Fitness>>
  <<Home> <About> <Programs> <Team> <Blog> <Contact Us> <Get Started>>>
 <<<Drop-Ins>
   <Bring your child in for a drop-in to get a workout>>
  <<<If you are visiting from out of town or traveling for club sports,
    make sure your child's routine is not disrupted. Bring them to our
    drop-in for a full session!> <<1:1 drop-in for
```

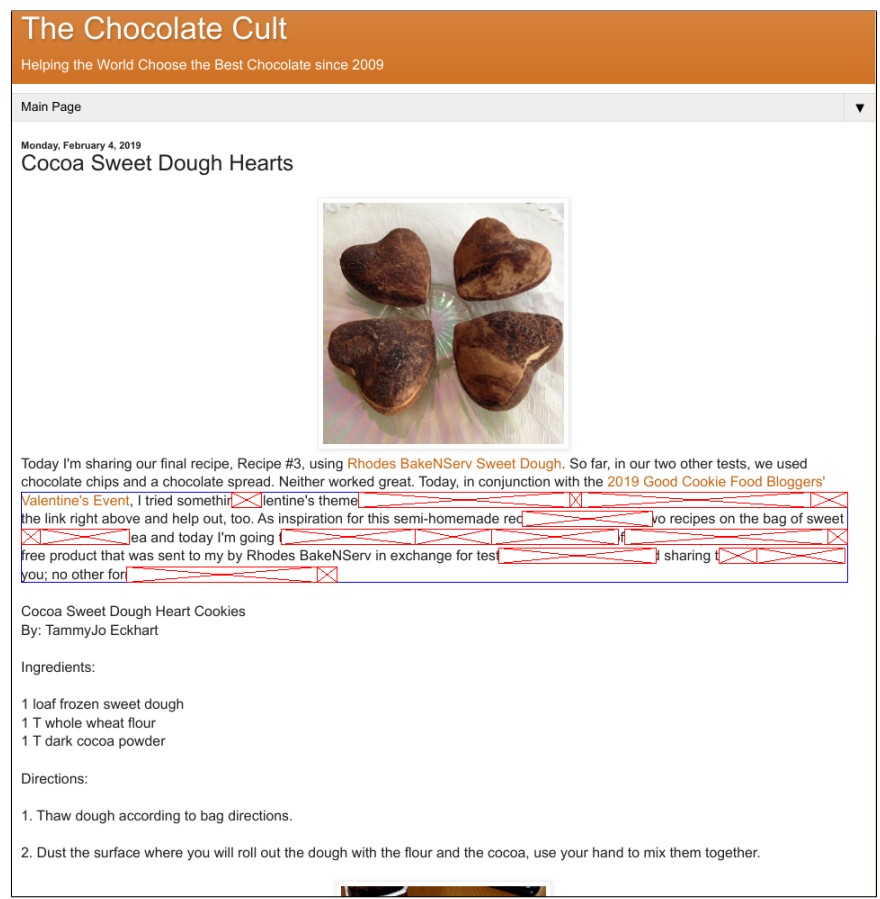

**Ground-truth Parse**

```
<, I tried something Valentine's themed. If you'd like to help
raise money for fighting children's cancer you can follow the link right
above and help out, too. As inspiration for this semi-homemade recipe,
I looked at the two recipes on the bag of sweet dough, I got an idea and
today I'm going to share with you how that worked out.
\xa0 I got the bag of Sweet Dough using a coupon for a free product
that was sent to my by Rhodes BakeNServ in exchange for testing out
their products and sharing the results with all of you; no other form of
compensation was received.>
```

**Predicted Parse**

```
<, I tried something Valentine's themed. If you'd like to help
out, I think you'd go right ahead and do a post. Click on the link right
above and help out, too. As inspiration for this semi-homemade recipe,
I've shared up two recipes on the bag of sweet dough. I got an idea and
today I'm going to share with you the second one.
Thank you for any of the amazing baking ideas plus this free product
that was sent to my by Rhodes BakeNServ in exchange for testing.
I'm really excited and sharing this recipe with all of you
```

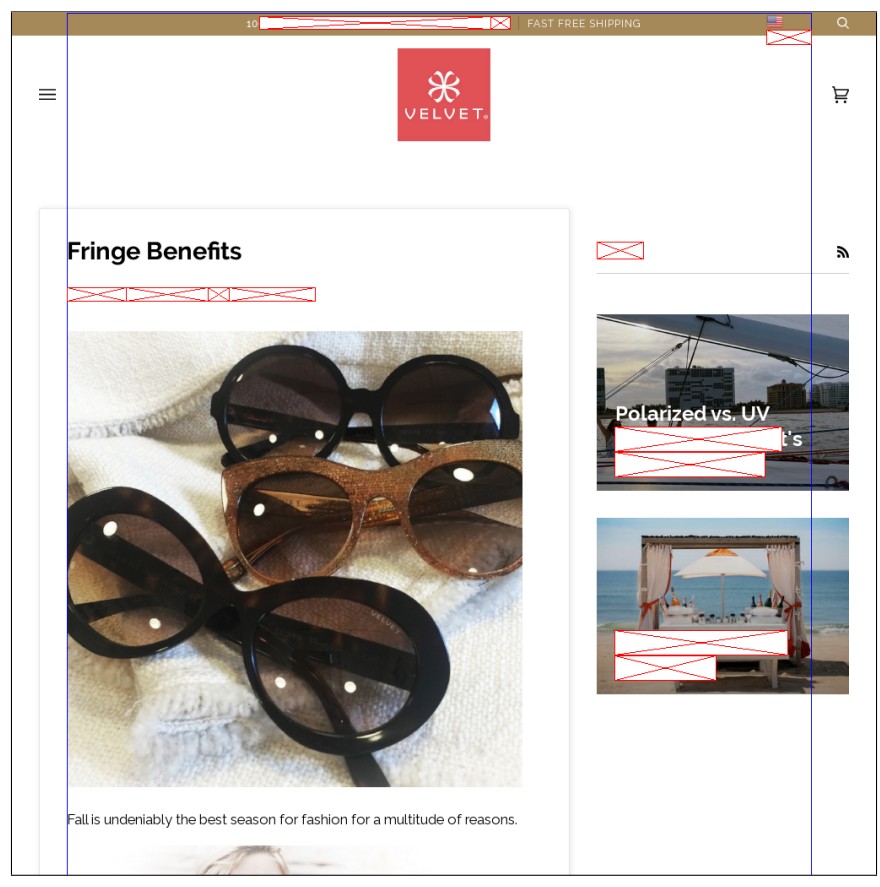

**Ground-truth Parse**

```
<<<100% FEMALE 100% UV PROTECTION SINCE 1999>
  <FAST FREE SHIPPING>>
 <img_alt=Velvet Eyewear>
 <<<<Fringe Benefits>
    <<Posted by> <Lindsay Sperin> <on> <August 19, 2016>>>
   <<img_src=img>
    <Fall is undeniably the best season for fashion
     for a multitude of reasons.>
    <img_src=img>>>
  <<NEWS>
   <<Polarized vs. UV Protection - What's The Difference?>
    <What's Hot in The Hamptons>>>>
 <<img_src=en-us img_alt=en> <English>>>
```

**Predicted Parse**

```
<<<10% OFF YOUR FIRST ORDER WITH CODE: FIRST10>
  <FAST FREE SHIPPING>>
 <img_alt=Velvet>
 <<<<Fringe Benefits>
    <<Posted by> <Velvet Fashion> <on> <October 1, 2018>>>
   <<Fall is undeniably the best season for fashion
     for a multitude of reasons.>
    <img_alt=Fringe Benefits>>>
  <<Search>
   <<Polarized vs. UV Protection: Velvet's Best Sunscreen>
    <The Best Sunblock Sunscreen>>>>
```

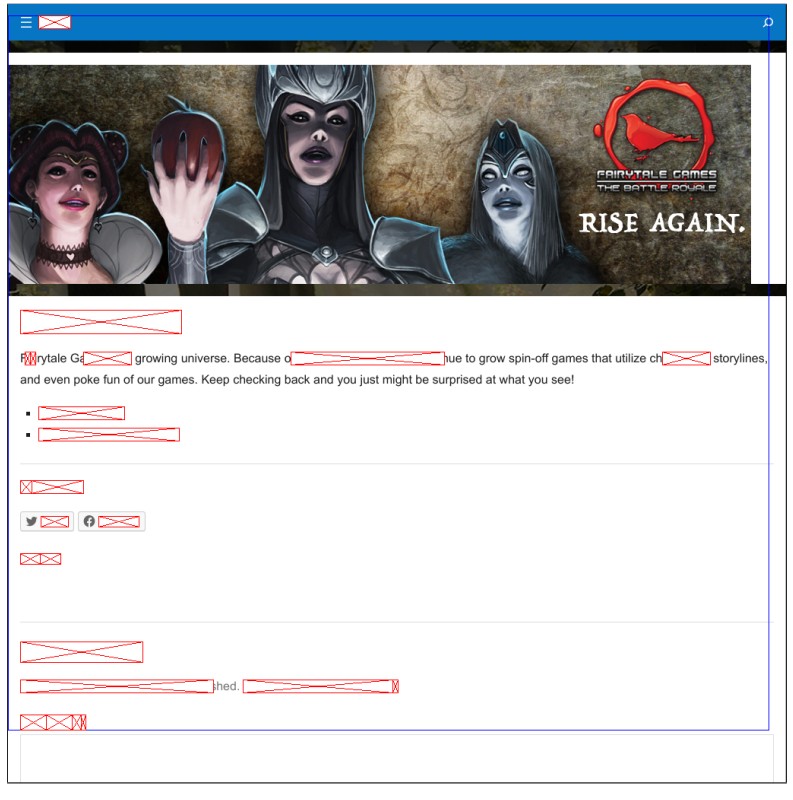

**Ground-truth Parse**

```
<<Menu>
 <img_src=ftg_webheader>
 <<<Spin-Off Games>
   <<Fairytale Games is a growing universe. Because of this, we have and
   will continue to grow spin-off games that utilize characters,
   storylines, and even poke fun of our games. Keep checking back and
   you just might be surprised at what you see!>
    <<Rumplestiltskin!>
     <Super Fairytale Fighters 2>>
    <<<Share this:>
      <<Twitter> <Facebook>>>
     <Loading...>>>>
  <<Leave a Reply>
   <<<Your email address will not be published.>
     <<Required fields are marked> <*>>>
     <<Comment> <*>>>>>>
```

**Predicted Parse**

```
<<Menu>
 <img_src=cropped-blogheader>
 <<<Fairytale Games>
   <<Fairytale Games is a growing universe. Because of this, we are
     excited to continue to grow spin-off games that utilize characters,
     storylines, and even poke fun of our games. Keep checking back and
     you just might be surprised at what you see!>
    <<Fairytale Games>
     <Fairytale Games on Steam>>
    <<<Share this:>
      <<Twitter> <Facebook>>>
     <Loading...>>>>
  <<Leave a Reply>
   <<<Your email address will not be published.>
     <<Required fields are marked
```

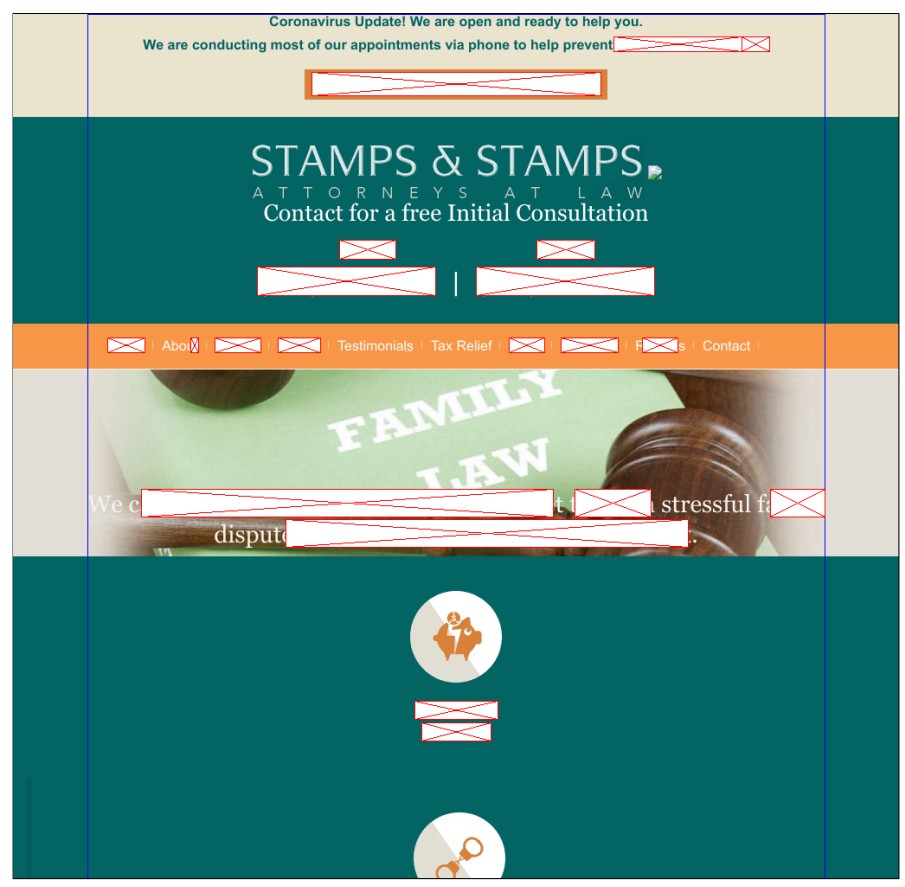

**Ground-truth Parse**

```
<<<<Coronavirus Update! We are open and ready to help you.>
   <We are conducting most of our appointments via phone to help prevent
    the spread of the virus.>>
  <Chapter 13 Coronavirus Update>>
 <<img_src=Logoo img_alt=Stamps & Stamps Attorneys At Law>
  <img_src=Phone>
  <Contact for a free Initial Consultation>
  <<Call Us> <(937) 247-6447>>
  <<Text Us> <(937) 265-6418>>>
 <<Home> <About> <Articles> <Videos> <Testimonials> <Tax Relief> <News>
  <Podcasts> <Rate Us> <Contact>>
 <<We can provide the guidance you need to get through stressful family>
  <disputes with your rights and interests intact.>>
 <<<img_src=Bankruptcy img_alt=Bankruptcy Overview>
   <<Bankruptcy> <Overview>>>
  <img_src=Criminal-Defense1
   img_alt=Criminal Defense & Traffic Offenses>>>
```

**Predicted Parse**

```
<<<<Coronavirus Update! We are open and ready to help you.>
   <We are conducting most of our appointments via phone to help prevent
    the spread of infection.>>
  <CLICK HERE FOR MORE INFO>>
 <<img_src=logo img_alt=Stamps & Stamps Attorneys At Law>
  <img_src=phone>
  <<<Call Us> <(904) 222-2222>>
   <<Text Us> <(904) 222-2222>>>>
 <<Home> <About> <Articles> <
```

