# OpenReview forum: "Pix2Struct: Screenshot Parsing as Pretraining for Visual Language Understanding"
_ICLR.cc/2023/Conference — Submitted to ICLR 2023_

### Official Review · Reviewer_dECs · 2022-10-25

**Confidence:** 3
**Correctness:** 3
**Technical Novelty And Significance:** 2
**Empirical Novelty And Significance:** 2
**Recommendation:** 5

**Clarity, Quality, Novelty And Reproducibility:**

 I have a major concern regarding the applicability and the reproducibility of the results. The base and large models contain many parameters and the pre-training requires very important resources which represent a huge financial and ecological cost. This could be an argument for not using this pre-training method in other applications. Moreover, few organisations or companies have the capacity or even the desire to devote so much money and resources to such pre-training, which makes the results non-reproducible.

**Details Of Ethics Concerns:**

Considering the ecological crisis, is it ethical to train a 1.3B parameters model on a 80M examples dataset during 170K steps on 128 Google Cloud TPUs ?

**Strength And Weaknesses:**

Although interesting and very well written, this paper does not make major contributions to the field.

The first contribution concerning the pre-training strategy is quite interesting. The authors propose a strategy based on web pages screenshots that allows to collect quickly and easily a very large amount of data. Moreover, these data can be quite varied and contain a large variety of elements, which allows to train a model that can work on many tasks.

The second contribution consists in adding modifications to the transformer inputs to handle variable aspect ratios and resolutions. This strategy, although interesting, brings little gain to the results. Moreover, it does not constitute a major contribution to the paper.

For the experiments, the following comments should be addressed.

1. The idea of simplifying the multi-modalities problem is interesting. However, it could have been nice to compare the method to a standard approach with a modality combination.
2. The input masking is done using crossed-out opaque bounding boxes. There is no justification about this choice. I wonder if this choice has an impact on the results compared to other approaches like removing the boxes or using white bounding boxes.
3. The authors claim that the changes they added to the standard ViT inputs provide major advantages in terms of robustness to extreme aspect ratios and on-the-fly changes to the sequence length. However, except for the results presented on Figure 4, there is no experiments showing these advantages.
4. It is not clear whether the modifications applied to the screenshots and HTML only consist in masking some texts or if other modifications are applied.
5. I would have appreciated to have the inference times of the base and large models.
6. The method of Wang et al (2021a) for the Screen2Words task could be more detailed.
7. It is not said on which task and dataset the ablation study on the inputs resolution is carried on.
8. 13 articles are cited as ArXiv preprints, please update the citations for articles that have been published.
inor comments:

1. In Table 2, it would make the table clearer to add a line with the metrics.
2. In Table 2, the value 160.4 should be highlighted instead of 145.0.
3. The values should be given with the same number of significative numbers:
    - 11.27 in AI2D
    - 145 in TextCaps
    - 40 and 81 in InfographiVQA
4. Figure 3 is a table, it should be renamed Table 3.
5. Typo:
    - Figure 4 caption: "Our variable-resolution inputs prevent**s**..."
6. In the appendix
    - In the fine-tuning section, it is said "Tables 4 and 4", it should be corrected to "Tables 4 and 5". By the way, tables 4 and 5 are quite similar, they could be merged into a single table.

**Summary Of The Paper:**

This paper presents Pix2Struct, a pretrained image-to-text model. The model is pretrained to parse masked screenshots of web pages and the authors show that it can be fine-tuned on multiple tasks containing visually-situated language (VQA, Image captioning, Infographic VQA). The fine-tuned models achieve state-of-the-art results in six tasks, while using only pixel-level inputs. The authors also introduce a variable-resolution input representation to prevent the distortion of input images.


**Summary Of The Review:**

Although the proposed method for pre-training a model from web page screenshots is interesting, the conditions of the experiment, with very large amounts of data and mobilizing unreasonable computational resources, do not allow to highlight its interest, its applicability and the reproducibility of the results.

---

> ### Author Response · Authors · 2022-11-19
> **Response to Official Review of Paper5452 by Reviewer dECs**
>
> Thank you for the review. We appreciate that you find the pre-training strategy interesting. We would like to clarify that this is in fact our primary contribution. We consider the contribution of using variable resolution as a simple “bugfix” to the standard ViT to make it robust to more scenarios.
>
> We also appreciate the detailed suggestions to formatting errors. They have been fixed in the latest revision. Please see responses to major points of discussion below:
>
> ### Comparing to modality combination
> We agree that it is important to compare traditional modality approaches, and we have done so extensively in the experiments. In the main results table, the first row corresponds to the state-of-the-art approaches, all of which are pipelines that combine different modalities. These approaches include pipelines that combine (1) OCR and images, (2) android view hierarchies and images, (3) diagram parses and images, and (4) table prediction and images. The difficulty of studying the cross-product of all systems and all tasks due to the specialization is a major motivation for exploring more general-purpose methods.
>
> ### Arbitrary choices during pretraining
> There are many arbitrary choices to be made in the formatting of the pretraining (e.g. blank vs crossed-out rendering of bounding boxes). We do not claim that these particular choices are uniquely useful beyond the requirement that they be unambiguous. It is possible that such choices matter, but we focused our resources on ablating the aspects of pretraining that we think are central to our research hypotheses.
>
> ### Modifications to the screenshot and HTML
> We modified the screenshot and HTML as was described in the paper (masking some text, drawing the bounding box around the region of interest, and compacting the HTML tree). Other unstated modifications were not applied.
>
> ### Inference times
> In the appendix, we have added a figure showing various inferences times for the different model sizes and different input resolutions for the DocVQA task. On a v3-8 Cloud TPU (the smallest v3 TPU that can be created) we observe ~62-255 documents/second for the base model and ~20-80 documents/second for the large model (depending on the input resolution).
>
> ### Computational costs
> Our model is not significantly larger than existing models. In fact, it is relatively small compared to other self-supervised models used in recent years. For example, GPT-3, a commonly used model in NLP research, consists of 176B parameters (compared our 282M and 1.3B). Our model variants are comparable in size to T5 variants which were trained three years ago.
>
> Additionally, as with existing pretrained models, spending resources upfront during general-purpose pretraining allows us to save on resources in the long run since this work can be reused for every new task. These benefits could be particularly pronounced since our method performs well across a variety of domains, each of which previously required domain-specific pretraining.
>
> Lastly, our method is effective even at smaller scales (282M parameters). The large model only serves as an indication of the benefits of further scaling. Finally, we have made our model checkpoints and code publicly available, which will allow others to build efficiently on the resources that we have spent. Both model variants can be fine-tuned on academic-scale hardware.

---

### Official Review · Reviewer_GpRK · 2022-10-26

**Confidence:** 5
**Correctness:** 4
**Technical Novelty And Significance:** 3
**Empirical Novelty And Significance:** 4
**Recommendation:** 6

**Clarity, Quality, Novelty And Reproducibility:**

The idea presented in this paper is novel and the quality of the proposed method is good.

I have concerns about this paper’s presentation and reproducibility. As mentioned before, many technical details are missing. The paper uses a lot of space to explain the downstream datasets and previous models for them, which seems to me unnecessary. I would suggest to use a figure for each dataset to explain the task and how Pix2Struct handles it, move the details to appendix, and use more space to describe the data processing or modeling of Pix2Struct in more detail.

This paper can be easily understood, but the model cannot be easily reproduced. I appreciate the authors’ effort for processing the web data and designing the model, which requires a lot of engineering. However, it is hard to follow this research if they cannot be made public.

**Strength And Weaknesses:**

Strength:

1. Pix2Struct uses a general-purpose pixel-to-text design, which simplifies the model architecture and can be easily applied to multiple domains.
2. The masked screenshot parsing objective is intuitive and effective. The pretraining data is easy to obtain from the web. The warmup stage with an image-to-text curriculum further improves the pretrained model.
3. The proposed fine-tuning strategy seamlessly integrate language and vision inputs by rendering language prompts on the image, without changing the architecture.
4. A single pretrained model achieves strong performance on multiple visual language understanding tasks from diverse domains.

Weakness:

1. The presentation of this paper is easy to follow, but not clear enough. Many important techniques are described at a high level, for example

    (1) How the proposed variable-resolution input representation is implemented, how to determine the number of patches for width and height, do small and large inputs have the same number of patches?

    (2) How the pretraining data is collected and processed, especially for the warmup stage.

    (3) How the masked parts are selected?

2. The authors do not mention if the data/code/model will be publicly available. It would be difficult for the community to reproduce or follow if they cannot be released.

3. Some of the designs lack empirical justification. For example, are there alternative ways to fine-tune the model, such as concatenating image and text? Does the location/size/font of where the prompt is rendered affect the performance?


**Summary Of The Paper:**

This paper proposes a pretrained image-to-text model for visual language understanding, which has a wide range of application in sources such as textbooks with diagrams, webpages with images and tables, and mobile apps with buttons and forms. The model has a simple and general architecture which only takes an image as the input. It is pretrained by parsing masked screenshots of webpages into simplified HTML, which subsumes common pretraining signals. This paper also proposes an integration of language and vision inputs for fine-tuning the model. The model achieves SOTA results in six out of nine tasks across four domains.


**Summary Of The Review:**

This paper proposes a simple pretraining framework for visual language understanding. The simplicity of the model, taking only image as the input, enables it to apply to diverse tasks and achieve supreme results. It is an exciting innovation in this area, and I believe it would greatly benefit the community if the data/model can be made public.

---

> ### Author Response · Authors · 2022-11-19
> **Response to Official Review of Paper5452 by Reviewer GpRK**
>
> Thank you for the review. Your major concern is on the replicability of our method, and we hope that is will be addressed by our open sourced code and checkpoints.
>
> Regarding our design choice of finetuning with the rendered prompt: we are drawing the closest analogy to what has been the standard approach in NLP, and we argue that this is in fact the simplest choice. When finetuning NLP models for tasks with prompts and documents, they are inserted into the same space to ensure similarity with what the transformer observed during pretraining. This allows off-the-shelf language models to be used directly without needing to pretrain a separate “prompt” channel. Similarly, we avoid needing to pretrain a specialized prompt channel—pix2struct was already pretrained to (amongst other things) reason about relationships between headers and main content.

---

### Official Review · Reviewer_Apqp · 2022-10-27

**Confidence:** 3
**Clarity, Quality, Novelty And Reproducibility:** The paper is clear and makes a novel …
**Correctness:** 3
**Technical Novelty And Significance:** 2
**Empirical Novelty And Significance:** 3
**Recommendation:** 6

**Strength And Weaknesses:**

The paper makes a welcomed and impressive empirical contribution, with several nice discussions on certain design choices. I will note 3 concerning points:

1. OCR or no OCR?

The paper is in line with the trend of replacing domain-specific components with a general pre-trained neural model, which is generally welcomed. However, in domains where OCR is cheap and adequate, I am not sure if it is worthwhile to spend so much compute on replacing the OCR with a pixel-only LM.

An ideal solution would be an adaptive model which is pre-trained on such general screenshot parsing task but when adapted to a downstream task, could be fine-tuned to accept domain-specific inputs when there are cheap and easy-to-use solutions.

2. Scaling up the model parameters seems to only bring marginal performance improvement.

Is there any observation about this? Is it because the pre-training task is too simple? It would also be good if the authors could provide evaluations on the pre-training task itself, using some simple metrics such as blue or exact match.

3. More details on how the pre-training data are extracted and cleaned should be provided.

The authors mentioned that the data are from C4. Does C4 already provide the cleaned data in HTML? My impression of C4 was that it was primarily a web-text corpus.

Statistics on the pre-training data should also be provided. E.g., what’s the lengths distribution of the output strings?



**Summary Of The Paper:**

The paper proposes screenshot parsing as a simple yet effective pre-training task for visually-situated language modeling (e.g., document QA). The model gets rid of the OCR and directly accepts an image (with rich layouts and rendered text) as input. The empirical results show that the pre-training outperforms prior pixel-only methods by a large margin.

**Summary Of The Review:**

Overall, the paper shows that screen parsing is an effective pre-training task for visually-situated language modeling. I am on the fence about the design choice of using a pure pixel-only transformer but that does not undermine the core argument for screen parsing.

---

> ### Author Response · Authors · 2022-11-19
> **Response to Official Review of Paper5452 by Reviewer Apqp**
>
> Thank you for the review. Suggested additions such as intrinsic evaluation of pretraining and information about sequence lengths have been added to the latest revision. Please see responses to major points of discussion below:
>
> ### OCR or no OCR?
> In addition to the discussion of this issue in the general response, we’d also like to mention that in many cases OCR is not cheap. OCR can be costly since it is also a visual model that itself is making increasing use of deep learning. In fact the authors of Donut (a major point of comparison in the paper) have pointed out that a key benefit of OCR-free models is that they’re often faster, since the cost of OCR can dominate the inference pipeline.
>
> An adaptive hybrid solution is a great idea that we are working on next but is out of scope for this paper. We believe that the current results showing what is possible with a purely-visual model will serve as a clean point of reference for such systems moving forward.
>
> ### Scale and pretraining corpus
> While the large model does show only modest improvement over the base model, we have reason to believe that this is for reasons somewhat orthogonal to our proposed method.
>
> First, due to resource constraints, the large model is pretrained with a smaller batch size and fewer training steps. It is known from T5 and RoBERTa that large batch sizes and many training steps contribute greatly to the quality of the pretrained model. The large model should be considered a best-effort analysis to give a hint at the potential benefits of scaling.
>
> Second, we also think the scaling benefits are hampered by the use of C4. We chose C4 for the sake of reproducibility, since it is a publicly available dataset that contains URLs and has a history of being used as a pretraining corpus. But as you correctly pointed out, C4 is typically used for its cleaned web-text corpus, and we are retrieving the original HTML based on the URLs. Since this was not the intended usage of C4, we think improving the quality and suitability of the pretraining corpus is important future work.

---

### Author Response · Authors · 2022-11-19
**General response to all reviewers**

Thank you for the thorough reviews. We include here responses to concerns that are common to all reviews.

The first common concern is the replicability of our results. We agree that this is crucial for progress in this area. We have open sourced our pretrained checkpoints (both base and large models), along with code that fully replicates all of our results across all nine tasks. We have added an anonymized link to the open-sourced code in the draft, which we will deanonymize after the review period.

The second common concern is whether consuming both text and vision inputs together via the pixel space is truly the best design choice. We’d like to clarify that we do not claim that pixel-only models are strictly better or that OCR components should be removed for all systems moving forward. However, we think that this is an exciting and plausible paradigm in the near future given the recent major advances in technology (specific reasons mentioned in the paper intro). Our work is an early data point towards addressing the research question of whether the benefits of self-supervised learning and end-to-end models outweigh the benefits of reusing existing tools such as OCR in a pipeline. Given that the overwhelming majority of prior work has leveraged OCR-based features, it seems necessary to advance OCR-free alternatives (as this paper does) in order to enable a clearer longer-term understanding around the proper role for OCR.

---

### Decision · Program_Chairs · 2023-01-20

**Decision:**

Reject

**Justification For Why Not Higher Score:**

While the all reviewers agree that the proposed vision–language pretraining strategy is quite interesting and thought to be useful, the central concern is the reproducibility issue because the method involves many algorithmic details as well as huge amount of web data and computational resource. The authors replies that the code and pretrained checkpoints are open sourced, but unfortunately, we cannot confirm them at the link added in the paper. Yes, we know it is a dummy for anonymization and maybe the code is really published somewhere, but we cannot check them without actively searching the web and breaking authors' anonymity, which is not allowed. (at least, the authors could have submitted a part of the code with anonymization as a supplementary material)

Related to this, one reviewer raised a concern regarding the presentation of the paper that many important techniques are described at a high level and unclear. It seems that this comment was not addressed very seriously (just saying 'we have open sourced the code') and its not clear how the paper will be improved. Since neither the code nor the better description can be verified, the AC believes that the reproducibility issue still remains. There was no strong support from the reviewers, so the AC would like to recommend rejection accordingly.

**Justification For Why Not Lower Score:**

N/A

**Metareview: Summary, Strengths And Weaknesses:**

Summary:
This paper proposes a pretrained image-to-text model for visual language understanding. The proposed model, Pix2Struct, is pretrained by learning to parse masked screenshots of web pages into simplified HTML. The model is OCR-free and directly accepts an image as input, which is thought to be computationally heavy but more general. Its fine-tuned model achieves SOTA results in six out of nine tasks across four domains.

Strengths:
1. The proposed pretraining strategy is intriguing and seems practically useful.
2. The model achieves good performance on wide range of downstream tasks.

Weaknesses:
1. Reproduciblity of the method is questionable mainly due to the lack of details in presentation. Also, it is still not certain whether the code is really available.
2. The method requires huge amount of data and computational resource, although this is a common problem for this kind of pretraining.